# Sex-specific single cell-level transcriptomic signatures of Rett syndrome disease progression
Osman Sharifi [1,2,3], Viktoria Haghani [1,2,3], Kari E. Neier[1,2,3], Keith J. Fraga[2,4], Ian Korf[2,4], Sophia M. Hakam[1,2,3], Gerald Quon [2,4], Nelson Johansen[2,4], Dag H. Yasui [1,2,3,5] & Janine M. LaSalle [1,2,3,5] ✉

Dominant X-linked diseases are uncommon due to female X chromosome inactivation (XCI). While random XCI usually protects females against X-linked mutations, Rett syndrome (RTT) is a female neurodevelopmental disorder caused by heterozygous *MECP2* mutation. After 6-18 months of typical neurodevelopment, RTT girls undergo a poorly understood regression. We performed longitudinal snRNA-seq on cerebral cortex in a construct-relevant *Mecp2e1* mutant mouse model of RTT, revealing transcriptional effects of cell type, mosaicism, and sex on progressive disease phenotypes. Across cell types, we observed sex differences in the number of differentially expressed genes (DEGs) with 6x more DEGs in mutant females than males. Unlike males, female DEGs emerged prior to symptoms, were enriched for homeostatic gene pathways in distinct cell types over time and correlated with disease phenotypes and human RTT cortical cell transcriptomes. Non-cell-autonomous effects were prominent and dynamic across disease progression of *Mecp2e1* mutant females, indicating that wild-type-expressing cells normalize transcriptional homeostasis. These results advance our understanding of RTT progression and treatment.

Rett syndrome is a neurodevelopmental disorder primarily affecting females and is characterized by a range of symptoms such as loss of speech, motor abnormalities, and developmental regression at about 6–18 months of age[1]. Rett syndrome most often occurs through spontaneous missense or truncation mutations, thereby predominantly affecting XX females[2]. Males with *MECP2* mutations that survive infancy are rare but can be observed in clinical phenotypes ranging from severe neonatal encephalopathy to cognitive impairment[3]. *MECP2* encodes the DNA binding protein, methyl CpG binding protein 2 (MeCP2), which is a critical regulator of neuronal gene expression in the brain. Among the two alternatively spliced *MECP2* transcripts, only the MeCP2e1 isoform contributes to RTT disease phenotypes[4]. However, most mouse studies of RTT utilize the exon 3-4 knockout model in *Mecp2-/y* males, which is a null model effective for studying MeCP2 function, but not a construct- or sex-relevant model for human RTT[5]. RTT females are heterozygous for *MECP2* (*MECP2-/+*) mutations and are therefore mosaic for both *MECP2* wild-type and mutant cells in the brain. Prior studies in Rett syndrome suggest potential non-cell-autonomous effects of MeCP2 deficiency on wild-type expressing cells in

the brain, but these effects have been poorly characterized at a cellular and molecular level[6–8]. RTT is characterized by a seemingly typical development in infancy, followed by progressive stages of regression in developmental milestones beginning around 6–18 months of age and lasting through early adulthood[1]. We have previously demonstrated that the *Mecp2e1* deficient mouse model of RTT, modeled after a human mutation, recapitulates the RTT-relevant extended period of disease symptom progression[4,9,10]. However, it is not known when and in which cell types the molecular changes responsible for disease progression occur in *MECP2* mutant females versus males.

To explore the effects of cellular mosaicism, sex, and cell type on the progression of disease in Rett syndrome, we employed single nuclei RNA-seq (snRNA-seq 5') analysis in the cerebral cortex of the *Mecp2e1* mutant mouse model. We examined the influence of sex, cell type, cellular mosaicism, and disease stage, correlated with progressive disease phenotypes using a systems-level perspective. These results demonstrate that MeCP2 deficiency in females results shows an inherently different disease progression at the cellular and molecular level compared to males, involving

[1]Medical Microbiology and Immunology, School of Medicine, University of California, Davis, CA, USA. [2]Genome Center, University of California, Davis, CA, USA. [3]MIND Institute, University of California, Davis, CA, USA. [4]Cellular and Molecular Biology, College of Biological Sciences, University of California, Davis, CA, USA. [5]These authors contributed equally: Dag H. Yasui, Janine M. LaSalle. ✉e-mail: jmlasalle@ucdavis.edu

stage-specific, non-cell-autonomous transcriptional changes to homeostatic gene pathways that correlate with disease phenotypes and stages.

## Results

### Experimental design to test longitudinal, cellular, and sex-specific transcriptional dysregulation in a symptomatically progressive mouse model of Rett syndrome

To identify sex, cell type, and disease stage specific transcriptional differences in *Mecp2e1* deficient mouse cortex, single nuclei RNA sequencing (snRNA-seq 5') analysis was performed to include the engineered mutation at the 5' translational start site of the *Mecp2e1* isoform[4]. Four longitudinal post-natal time points were chosen to correspond to pre-symptomatic (PND 30), disease onset (PND 60) and late disease stages (PND 120 for *Mecp2e1*[-/y] males, PND 150 for *Mecp2e1*[-/+] females) compared to sex-matched wild-type (WT) littermates[9,10] (Fig. 1a). A total of 28 cortex samples were used, including two per timepoint and genotype for males (PND30, 60, 120) and two per genotype for females at PND30 and 60, but four replicates per genotype were used in females at P150 in order to study mosaicism (Table 1).

Cortical nuclei were assigned to 14 different cell types based on 3,000 cell marker genes from the Allen brain atlas cortex transcriptomics data[11]. 93,798 cells from both sexes, four genotypes and four time points were all clustered unsupervised (Fig. 1b). Four excitatory neuron cell types were identified, corresponding with cortical layers 2 to 6 (L2-6), as well as five inhibitory cell types (Pvalb, Vip, Sst, Sncg, Lamp5) and five non-neuronal cell types (pericytes, endothelial, oligodendrocytes, astrocytes, non-neuronal including microglia). Unbiased marker genes for all 14 cell types were identified, supporting the distinction of our candidate cell types (Fig. 1c).

Five separate hypotheses were tested, comparing cells across different genotypes and expression phenotypes (mutant vs wild-type-expressing cells within females). In addition to comparing cells from *Mecp2e1*[-/y] to *Mecp2e1*[+/y] (experiment 1) and *Mecp2e1*[-/+] to *Mecp2e1*[+/+] (experiment 2), wild-type *Mecp2e1* expressing cells from the *Mecp2e1*[-/+] females were compared to the wild-type expressing cells from the *Mecp2e1*[+/+] (experiment 3) and mutant *Mecp2e1* expressing cells from the *Mecp2e1*[-/+] females were compared to either wild-type expressing cells from the *Mecp2e1*[+/+] (experiment 4) or wild-type expressing cells within *Mecp2e1*[-/+] females (experiment 5) to test for cell non-autonomous effects (Fig. 1d).

### Sexually dimorphic trajectories of transcriptional dysregulation across cortical cell types

To accurately characterize alterations in gene transcript abundance, four computational methods for identifying differentially expressed genes (DEGs) from single nucleus RNA sequencing (snRNA-seq) data were evaluated with single-cell data sets (Limma-VoomCC, Limma, EdgeR, and DESeq2) with partial overlap (Supplementary Fig. 1a). An overlap of DEGs between EdgeR and Limma-VoomCC showed that Limma-VoomCC identifies mostly the same DEGs, but EdgeR identifies many more DEGs that likely include many false positives, as highlighted by Squair et al.[12] (Supplementary Fig. 1b). Ultimately, Limma-Voom Consensus Correlation (Limma-VoomCC) was selected for DEG analysis based on the ability to reveal high expressing DEGs amongst diverse gene transcripts expressed[13]. Further, Limma-VoomCC controlled for the inter-correlations of cells from the same animals[14,15]. Overall, Limma-VoomCC analyses of all cell types in experiments 1 and 2 revealed a total of 1440 significant DEGs after adjusting for false discovery (Table 1). In males from experiment 1, 165 or 85% showed higher and 30 or 15% showed lower transcript levels in *Mecp2e1* mutant cortical cells compared to wild-type controls across the three time points, with fold changes ranging from a low of −1.99 for *Sst* to a high of +2.31 for *Cst3*. In females from experiment 2, 282 or 22% showed higher and 959 or 77% showed lower transcript levels in *Mecp2e1* mutant cortical cells compared to wild-type controls across the three time points, with fold changes ranging from a low of −2.69 for *Snhg11* to a high of +3.47 for *Ay036118* (Supplementary Data 1). DESingle was also used as a

complementary approach to identify lower confidence DEGs for transcripts expressed at low levels (Supplementary Fig. 2). To ensure that DEGs detected were not due to changes in cell types, we examined cell proportions which did not show changes over time (Supplementary Fig. 3). Cell clustering based on cell type, time point, sex and *Mecp2e1* genotype did not show evidence of batch effects (Supplementary Fig. 4a–d). Furthermore, an analysis of the top high and low expressing genes showed that brain samples from replicate mice were consistent (Supplementary Fig. 5).

Analysis of DEGs by Limma-VoomCC, DEsingle, and EdgeR revealed that cell type transcriptional changes associated with *Mecp2e1* deficiency were markedly different by sex and disease stage in multiple cortical cell types and that these differences were consistent across bioinformatic methods (Fig. 2, Supplementary Fig. 2a–f). At the pre-symptomatic stage, *Mecp2e1*[-/y] male P30 from experiment 1 cortical cells had only 9 DEGs compared with wild-type, including 3 DEGs in L2/3 neurons, 4 DEGs in L4 neurons (including immediate early genes *Arc* and *Junb*), and 1 DEG (*AC149090.1*) in Lamp5 and Vip neurons (Fig. 2a). In contrast, *Mecp2e1*[-/+] female from experiment 2 single cortical cells showed the strongest transcriptional dysregulation, for a total of 1215 DEGs at P30 (Limma-VoomCC). Interestingly, *Mecp2e1*[-/+] female Pvalb DEGs at P30 had a significant (*p-value* ≤ 0.00075) enrichment of imprinted genes, including *Meg3, Xist, Gnas, Kcnq1ot1, Np1l5, Ntm, Peg3* and *Snrpn* (Fig. 2b), a result that was not observed in males.

At the disease onset P60 timepoint, 73 DEGs were identified in *Mecp2e1*[-/y] males, with 56 DEGs in L2/3 neurons, but also including 7 DEGs in astrocytes, 4 DEGs in L4 and 6 in Scng neurons and 1 DEG in L6 and Sst neurons (Fig. 2c). *Mecp2e1*[-/+] female cortical cells had 47 DEGs, with 46 in L6 excitatory neurons and 1 DEG in Sncg inhibitory neurons (Fig. 2d). Further, *Mecp2e1*[-/+] female DEGs at P60 included *AY036118* (+3.47-fold change), *Ptprd, Edil3, Ptgds, Plp1, Atp6v0b, Kcn11ot1, Gria2, Nrxn1, Arpp21, Snhg11*. *Mecp2e1*[-/+] females had 3 DEGs with 2 in VIP inhibitory neurons and 1 DEG in Pvalb inhibitory neurons.

By the late disease P150 time point, only VIP interneurons contained DEGs in *Mecp2e1*[-/+] cortical cell types, including long non-coding RNAs *Snhg11* (p value = 0.0043) and *Meg3* (*p-value* = 0.017). Remarkably, *Mecp2e1*[-/+] female cortical cells were most transcriptionally dysregulated prior to the onset of symptoms, as the number of DEGs decreased in number as disease symptoms progressed (Fig. 2d). Overall, *Mecp2e1*[-/y] male DEGs increased in number with disease progression, but *Mecp2e1*[-/y] male cortical cell types had only 199 DEGs across all three time points, which was only 16.3% of the total *Mecp2e1*[-/+] female DEGs (Fig. 2c, d).

To identify enriched functional pathways connecting RTT transcriptional progression, Kyoto Encyclopedia of Genes and Genomes (KEGG) analysis was performed using the DEGs (Limma-VoomCC, *p-value* ≤ 0.05) from each cell type. KEGG pathways that were persistent over P30, P60, and P120 in *Mecp2e1*[-/y] male cortical cells or P30, P60 and P150 in *Mecp2e1*[-/+] female cortical cells are shown (Fig. 2e, f, Supplementary Data 2). Distinctly different pathway dysregulation was observed between *Mecp2e1*[-/+] females and *Mecp2e1*[-/y] male cortical cells by two key metrics. First, *Mecp2e1*[-/+] cortical cell DEGs were enriched for 18 different pathways consistently across time points, compared to only two in *Mecp2e1*[-/y] males, of which only gastric acid secretion overlaps with *Mecp2e1*[-/+] pathways. Second, specifically in pre-symptomatic P30 *Mecp2e1*[-/+] females, Pvalb and Sst neurons shared 14 enriched pathways including synaptic vesicle cycle, retrograde endocannabinoid signaling, oxytocin signaling, morphine addiction, long-term potentiation, insulin secretion, glutamatergic synapse, gastric acid secretion, dopaminergic synapse, circadian entrainment, cAMP signaling pathway, amphetamine addiction, Alzheimer's disease, and aldosterone synthesis and secretion (Fig. 2f) compared to only the gastric acid secretion pathway at the disease onset P60 time point and late disease P120 time point in *Mecp2e1*[-/y] males (Fig. 2e). Interestingly, by symptom onset at P60 in *Mecp2e1*[-/+] females, 6 dysregulated KEGG pathways including nicotine addiction, long-term potentiation, insulin secretion, glutamatergic synapse, cAMP signaling, and amphetamine addiction (found in P30 Pvalb and Sst neurons) were distinctly significantly enriched in L5 and L6 excitatory

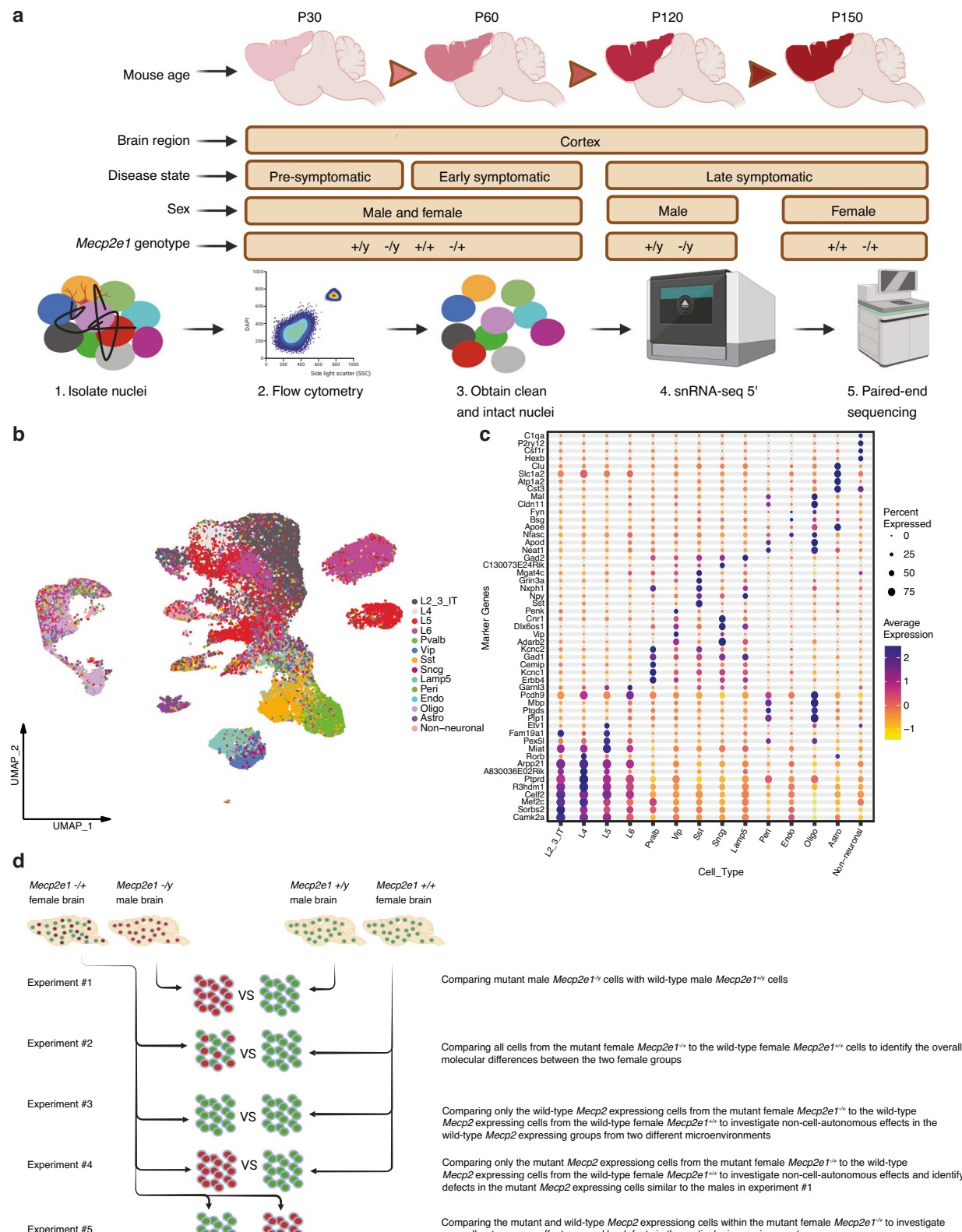

**Fig. 1 | A scheme showing the overall mouse study design. a** Cortical samples were collected from postnatal mice at four different timepoints corresponding to three different disease stages (n = 28). Four different Mecp2e1 genotypes were considered that include both sexes. **b** UMAP of the unsupervised clustering of cell types (n = 93,798 cells post QC) identified. Cell type labels were transferred from[39] Yao et al. 2021. **c** Top gene markers for each cell type are shown on y-axis. The color refers to the average expression of genes in a cell type and the percent expressed describes the percentage of cells within a cell type that express each gene marker. **d** Design of computational experiments comparing mutant to WT cells from mice of both sexes. Experiments 3 to 5 are comparing subtypes of cells in females due to X chromosome inactivation to examine potential non-cell-autonomous effects of *Mecp2e1* mutation. Figure 1a, d is made using Biorender.

**Table 1 | Summary of LimmaVoom DEG numbers resulting from the parsing of mutant- and WT-expressing cell type categories in all comparison experiments**

| Exp # | Cell type | #UP DEGs | #DOWN DEGs | Time point | Animal (n) | Total sig DEGs | Notes |
|---|---|---|---|---|---|---|---|
| 1 | Glutamatergic | 7 | 0 | P30 | 4 | 7 | Comparing all cortical glutamatergic cells between mutant male *Mecp2e1-/y* to WT male *Mecp2e1+/y* |
| 1 | GABAergic | 2 | 0 | P30 | 4 | 2 | Comparing all cortical GABAergic cells between mutant male *Mecp2e1-/y* to WT male *Mecp2e1+/y* |
| 1 | Non-neuronal | 0 | 0 | P30 | 4 | 0 | Comparing all cortical non-neuronal cells between mutant male *Mecp2e1-/y* to WT male *Mecp2e1+/y* |
| 1 | Glutamatergic | 0 | 27 | P60 | 4 | 27 | Comparing all cortical glutamatergic cells between mutant male *Mecp2e1-/y* to WT male *Mecp2e1+/y* |
| 1 | GABAergic | 4 | 1 | P60 | 4 | 5 | Comparing all cortical GABAergic cells between mutant male *Mecp2e1-/y* to WT male *Mecp2e1+/y* |
| 1 | Non-neuronal | 2 | 1 | P60 | 4 | 3 | Comparing all cortical non-neuronal cells between mutant male *Mecp2e1-/y* to WT male *Mecp2e1+/y* |
| 1 | Glutamatergic | 21 | 0 | P120 | 4 | 21 | Comparing all cortical glutamatergic cells between mutant male *Mecp2e1-/y* to WT male *Mecp2e1+/y* |
| 1 | GABAergic | 121 | 1 | P120 | 4 | 126 | Comparing all cortical GABAergic cells between mutant male *Mecp2e1-/y* to WT male *Mecp2e1+/y* |
| 1 | Non-neuronal | 8 | 0 | P120 | 4 | 8 | Comparing all cortical non-neuronal cells between mutant male *Mecp2e1-/y* to WT male *Mecp2e1+/y* |
| **1** | **Total** | **165** | **30** | | | **195** | |
| 2 | Glutamatergic | 31 | 913 | P30 | 4 | 944 | Comparing all cortical glutamatergic cells between mutant female *Mecp2e1-/+* to WT female *Mecp2e1+/+* |
| 2 | GABAergic | 236 | 35 | P30 | 4 | 271 | Comparing all cortical GABAergic cells between mutant female *Mecp2e1-/+* to WT female *Mecp2e1+/+* |
| 2 | Non-neuronal | 0 | 0 | P30 | 4 | 0 | Comparing all cortical non-neuronal cells between mutant female *Mecp2e1-/+* to WT female *Mecp2e1+/+* |
| 2 | Glutamatergic | 13 | 11 | P60 | 4 | 24 | Comparing all cortical glutamatergic cells between mutant female *Mecp2e1-/+* to WT female *Mecp2e1+/+* |
| 2 | GABAergic | 0 | 0 | P60 | 4 | 0 | Comparing all cortical GABAergic cells between mutant female *Mecp2e1-/+* to WT female *Mecp2e1+/+* |
| 2 | Non-neuronal | 0 | 0 | P60 | 4 | 0 | Comparing all cortical non-neuronal cells between mutant female *Mecp2e1-/+* to WT female *Mecp2e1+/+* |
| 2 | Glutamatergic | 0 | 0 | P150 | 8 | 0 | Comparing all cortical glutamatergic cells between mutant female *Mecp2e1-/+* to WT female *Mecp2e1+/+* |
| 2 | GABAergic | 2 | 0 | P150 | 8 | 2 | Comparing all cortical GABAergic cells between mutant female *Mecp2e1-/+* to WT female *Mecp2e1+/+* |
| 2 | Non-neuronal | 0 | 0 | P150 | 8 | 0 | Comparing all cortical non-neuronal cells between mutant female *Mecp2e1-/+* to WT female *Mecp2e1+/+* |
| **2** | **Total** | **282** | **959** | | | **1241** | |
| 3 | Glutamatergic | 17 | 782 | P30 | 4 | 799 | Comparing WT-expressing glutamatergic cells from mutant female *Mecp2e1-/+* to WT-expressing cells from WT female *Mecp2e1+/+* |
| 3 | GABAergic | 2 | 393 | P30 | 4 | 395 | Comparing WT-expressing GABAergic cells from mutant female *Mecp2e1-/+* to WT-expressing cells from WT female *Mecp2e1+/+* |
| 3 | Glutamatergic | 835 | 5 | P60 | 4 | 840 | Comparing WT-expressing glutamatergic cells from mutant female *Mecp2e1-/+* to WT-expressing cells from WT female *Mecp2e1+/+* |
| 3 | GABAergic | 5 | 0 | P60 | 4 | 5 | Comparing WT-expressing GABAergic cells from mutant female *Mecp2e1-/+* to WT-expressing cells from WT female *Mecp2e1+/+* |
| 3 | Glutamatergic | 2 | 41 | P150 | 8 | 43 | Comparing WT-expressing glutamatergic cells from mutant female *Mecp2e1-/+* to WT-expressing cells from WT female *Mecp2e1+/+* |
| 3 | GABAergic | 1 | 43 | P150 | 8 | 44 | Comparing WT-expressing GABAergic cells from mutant female *Mecp2e1-/+* to WT-expressing cells from WT female *Mecp2e1+/+* |
| **3** | **Total** | **862** | **1264** | | | **2126** | |
| 4 | Glutamatergic | 3 | 574 | P30 | 4 | 577 | Comparing mutant-expressing glutamatergic cells from female *Mecp2e1-/+* to WT-expressing cells from female WT *Mecp2e1+/+* |
| 4 | GABAergic | 9 | 984 | P30 | 4 | 993 | Comparing mutant-expressing GABAergic cells from female *Mecp2e1-/+* to WT-expressing cells from female WT *Mecp2e1+/+* |
| 4 | Glutamatergic | 1251 | 8 | P60 | 4 | 1259 | Comparing mutant-expressing glutamatergic cells from female *Mecp2e1-/+* to WT-expressing cells from female WT *Mecp2e1+/+* |
| 4 | GABAergic | 0 | 0 | P60 | 4 | 0 | Comparing mutant-expressing GABAergic cells from female *Mecp2e1-/+* to WT-expressing cells from female WT *Mecp2e1+/+* |

**Table 1 (continued) | Summary of LimmaVoom DEG numbers resulting from the parsing of mutant- and WT-expressing cell type categories in all comparison experiments**

| Exp # | Cell type | #UP DEGs | #DOWN DEGs | Time point | Animal (n) | Total sig DEGs | Notes |
|---|---|---|---|---|---|---|---|
| 4 | Glutamatergic | 4 | 8 | P150 | 8 | 12 | Comparing mutant-expressing glutamatergic cells from female *Mecp2e1-/-* to WT-expressing cells from female WT *Mecp2+/+* |
| 4 | GABAergic | 0 | 4 | P150 | 8 | 4 | Comparing mutant-expressing GABAergic cells from female *Mecp2e1-/-* to WT-expressing cells from female WT *Mecp2+/+* |
| **4** | **Total** | 1267 | 1578 | 0 | 0 | 2845 | |
| 5 | Glutamatergic | 0 | 0 | P30 | 2 | 0 | Comparing mutant-expressing glutamatergic cells from female *Mecp2e1-/+* to WT-expressing cells from female *Mecp2e1-/+* |
| 5 | GABAergic | 0 | 0 | P30 | 2 | 0 | Comparing mutant-expressing GABAergic cells from female *Mecp2e1-/+* to WT-expressing cells from female *Mecp2e1-/+* |
| 5 | Glutamatergic | 3 | 0 | P60 | 2 | 3 | Comparing mutant-expressing glutamatergic cells from female *Mecp2e1-/+* to WT-expressing cells from female *Mecp2e1-/+* |
| 5 | GABAergic | 0 | 0 | P60 | 2 | 0 | Comparing mutant-expressing GABAergic cells from female *Mecp2e1-/+* to WT-expressing cells from female *Mecp2e1-/+* |
| 5 | Glutamatergic | 6 | 0 | P150 | 4 | 6 | Comparing mutant-expressing glutamatergic cells from female *Mecp2e1-/+* to WT-expressing cells from female *Mecp2e1-/+* |
| 5 | GABAergic | 1 | 0 | P150 | 4 | 1 | Comparing mutant-expressing GABAergic cells from female *Mecp2e1-/+* to WT-expressing cells from female *Mecp2e1-/+* |
| **5** | **Total** | 10 | 10 | 0 | 0 | 10 | |

neurons (Fig. 2f). At the late disease P150 timepoint, *Mecp2e1*[-/+] female cortical astrocytes remarkably were significantly enriched for 15 out of the 18 total convergent KEGG pathways (Fig. 2f). While some of the reduced KEGG pathway enrichment in *Mecp2e1* deficient males compared to females could be due to fewer DEGs observed overall and especially at the pre-symptomatic stage in *Mecp2e1*[-/y] cortical cells, the significant enrichment of ubiquitin mediated proteolysis specifically at *Mecp2e1*[-/y] P30 when DEGs were fewest (Fig. 2e) suggests that DEG number is less important than the specificity of gene pathways dysregulated in the male *Mecp2e1*[-/y] mouse model. We also performed an enrichment analysis for DEGs based on gene length, but did not find evidence to support the previously reported repression of long genes in *Mecp2* deficient neurons of either sex[16] (Supplementary Fig. 6).

## Co-expression networks of dysregulated genes within cortical cell types correlate with *Mecp2e1* genotype, time point, sex, body weight, and disease score

To complement the DEG analysis, we performed a systems-biology based approach, High-Definition Weighted Gene Co-expression Network Analysis (hdWGCNA) which is specifically designed for analysis of high dimensional data such as snRNA-seq[17–19] (Fig. 3). hdWGCNA groups genes that are co-expressed together into colored modules based on scale-free topology[17,19] and was used to define nine distinct modules based on co-expression within a network built from transcriptomes of all detected genes from all cell types and experimental conditions. Genes in each module were compared to cell type marker genes to identify modules that uniquely correlate with phenotype (Supplementary Fig. 7). In co-expression network analysis, we focus on the hub genes, those which are highly connected within each module. Therefore, we determined the eigengene-based connectivity, also known as kME, of each gene. The top 10 ranked co-expressed hub genes were identified per module (Fig. 3a) and expression enrichment for each cortical cell type was determined, which was distinct from cell type markers (Fig. 3b). The blue hdWGCNA module corresponded to genes enriched in oligodendrocytes, while the magenta module genes were enriched in L5 and L6 neurons, with *Sez6* and *Nrp1* as hub genes. The brown module included *Grin2a*, *Grin2b*, and *Camk2a* and the black module included *Slit3* and *Gabrb3* enriched in excitatory neurons (L2-3, L4, L5, L6) showed similar cellular patterns of expression. The green module genes like *Grik1* and *Adarb2* were most highly expressed in inhibitory neurons. In contrast, the turquoise, pink, red and yellow modules are more cell type independent, being enriched in all neuronal subtypes (Fig. 3b).

To explore the relationship between cortical co-expression gene networks and disease progression in Rett syndrome, the eigengene value of each sample within each module was correlated with the body weight, disease score, genotype, sex, and disease time point of each mouse. Eigengene values were calculated for all cortical cells, as well as each cell type individually, so that correlations with each variable of interest could be examined for each cell type (Fig. 3c, Supplementary Fig. 7). While the genes within each module partially overlapped with those that served as cell type markers, the genes within modules were independent from those that defined cell type specificity (Supplementary Fig. 7). For all cell types, 6 out of the 9 gene modules significantly correlated with all phenotypes and experimental variables, and all modules showed at least one significant correlation (Fig. 3c, top row). Yet, certain gene set modules such as green correlated with phenotypes in L4, L6 and Sst neurons while the turquoise and blue modules correlated most strongly with phenotypes in all neuronal subtypes (Fig. 3c). Interestingly, magenta module genes only correlated with genotype in neurons. Astrocytes were distinct in that only blue and yellow modules correlated with both disease score and genotype. While most module-genotype correlations were positive (red), meaning that co-expressed genes in these modules were upregulated in mutant animals, the blue module uniquely was inversely correlated (blue), representing downregulated genes. Interestingly, module-sex associations were frequent but sometimes were absent in specific cell types or time points with strong module-genotype correlations, such as the blue module in L6. Pvalb, and Sst neurons (Fig. 3c).

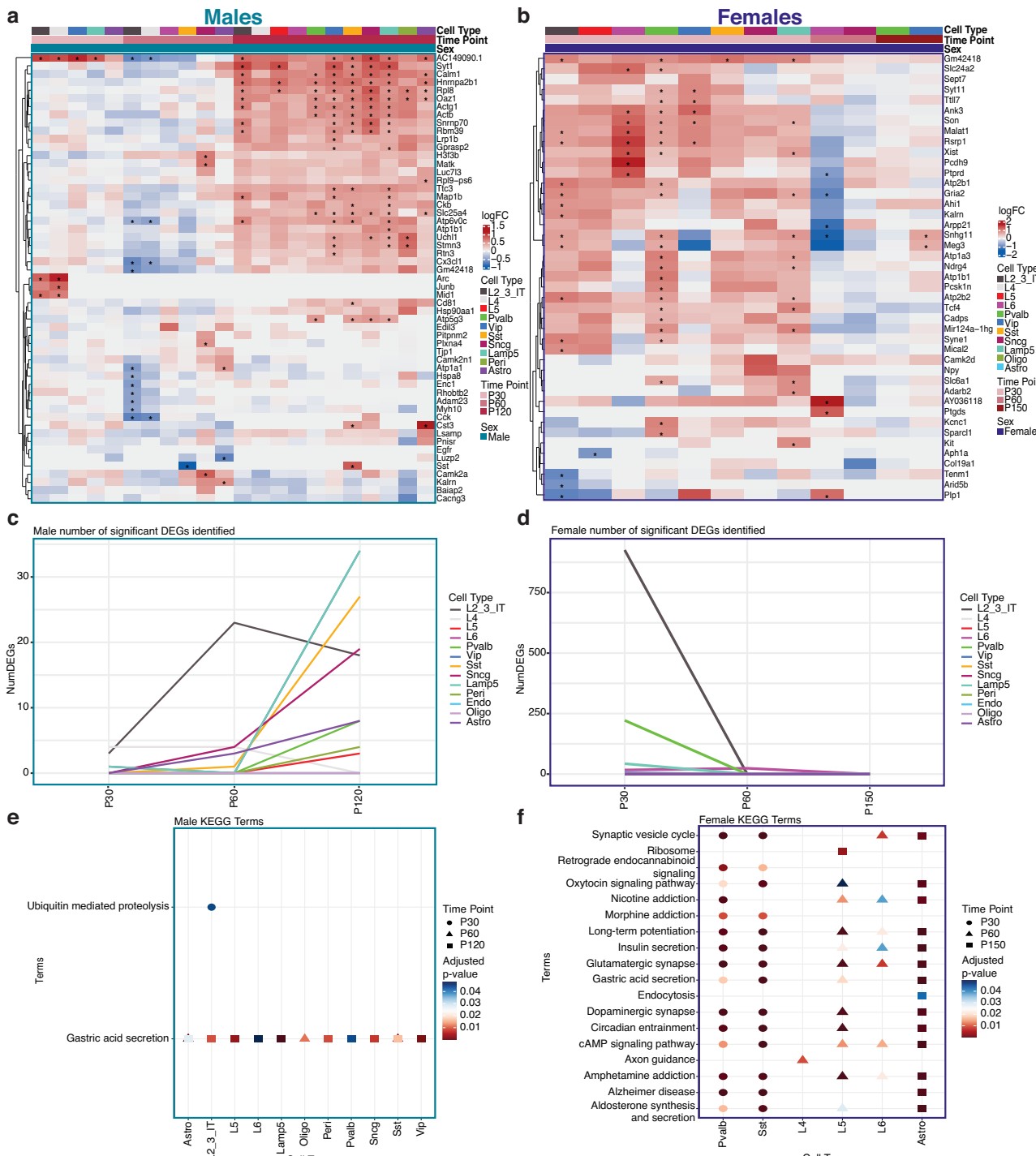

**Fig. 2 | Sexually-dimorphic dynamic patterns of DEGs and KEGG pathway terms across time and cell type. a** Heatmap of top 5 differentially expressed genes (DEGs) based on the lowest adjusted p-value ≤ 0.05 comparing male *Mecp2e1⁻/ʸ* and *Mecp2e1⁺/ʸ* cortical cells across timepoints (experiment 1). **b** Heatmap of top 5 DEGs comparing female *Mecp2e1⁻/⁺* and *Mecp2e1⁺/⁺* cortical cells across timepoints (experiment 2). *indicates adjusted p-value ≤ 0.05 (corrected via Benjamini and Hochberg method). **c, d** Number of DEGs over time at adjusted p-value ≤ 0.05 for experiments 1 and 2, respectively. **e, f** Dot plots showing the KEGG pathway terms for DEGs (adjusted p-value ≤ 0.1) from each cell type, selected for terms that are persistent over time for experiments 1 and 2, respectively.

Further, genes from each module were tested for KEGG pathway enrichment to identify phenotype correlated dysregulated pathways. Many RTT disease progression relevant pathways such as glutamatergic synapse, GABAergic synapse, circadian rhythm and axon guidance were identified (Supplementary Fig. 8). KEGG analysis for the turquoise module showed enrichment in neurological pathways such as Alzheimer disease and metabolic pathways such as choline metabolism (Supplementary Fig. 8).

## X chromosome expression mosaicism in female cortical cell populations reveals dynamic non-cell-autonomous transcriptional homeostasis

To examine non-cell-autonomous effects, we considered all *Mecp2* expressing cells within *Mecp2e1⁻/⁺* mosaic female cortical cells. Within *Mecp2e1⁻/⁺* female snRNA-seq data, we identified 1,146 *Mecp2* expressing cells, of which 607 could be genotyped as WT-expressing and 539 were

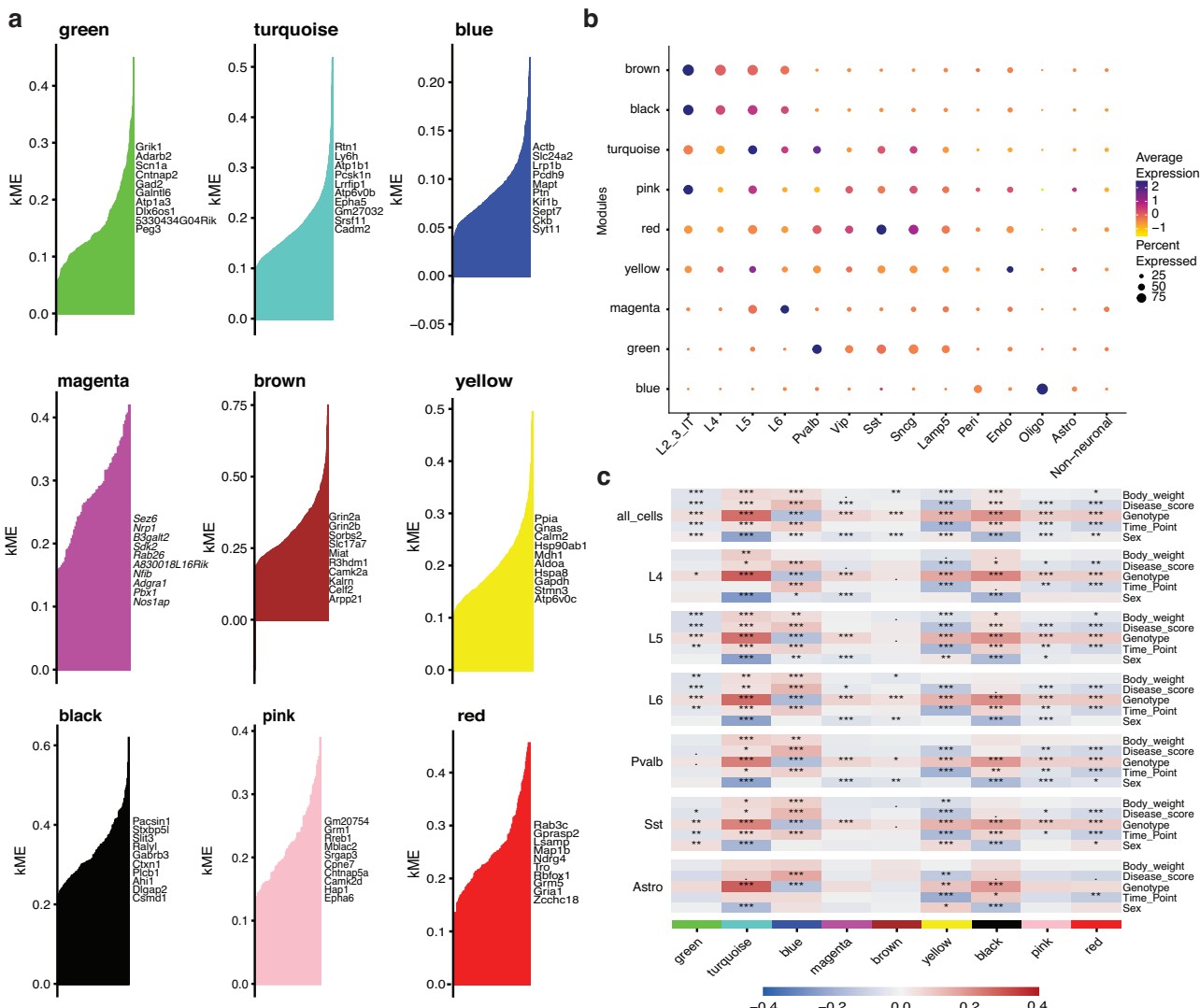

**Fig. 3 | hdWGCNA identifies co-expression networks for each cell type in the mouse cortex that correlated with *Mecp2e1* genotype, disease phenotypes, and sex. a** Top 10 hub genes identified for each of the 9 modules generated by hdWGCNA on entire snRNA-seq dataset, identified by color. The x-axis are all the genes in each module and the y-axis is the corresponding kME value. **b** Dot plot of the average gene expression of the top 10 hub genes in each module generated for each cortical cell type. **c** A heat map of correlations between experimental phenotypes and variables (body weight, disease score, genotype, time point, sex) and averaged gene expression (eigenmode value) for each cell type (cell types not shown are in Supplementary Fig. 7a). *, **, *** indicates FDR-corrected p-value ≤ 0.05, 0.01, and 0.001, respectively. The color bar shows the Pearson correlation coefficient.

expressing the *Mecp2e1* mutation (Supplementary Fig. 9). These cells were clustered based on expression and thus twelve different cell types were identified (Fig. 4a). To reduce the impact of lower cell counts on DEG calling following parsing, we further grouped the *Mecp2* expressing cells into two broad categories of GABAergic neurons and glutamatergic neurons (Fig. 4b). A summary of the number of DEGs in each of these three broader cell type categories (glutamatergic, GABAergic, non-neuronal for each of the five experimental comparisons (Fig. 1d) is shown in Table 1. Limma-Voom was used for DEG calling of experiments 3, 4 and 5. As expected based on random XCI, both cell populations (Mecp2_MUT and Mecp2_WT), were randomly represented in all cell types (Fig. 4a–c). Cells from the males were also parsed, showing 184 Mecp2_MUT in the *Mecp2e1*$^{-/y}$ and 175 Mecp2_WT in the *Mecp2e1*$^{+/y}$ (Supplementary Fig. 9).

In order to examine non-cell-autonomous effects over disease progression in a cell category specific manner, we followed the third experiment design (Fig. 1d) and compared the WT-expressing cells from the *Mecp2e1*$^{-/+}$ mosaic females to the WT expressing cells from the *Mecp2e1*$^{+/+}$ females (Fig. 4d, Table 1). At P30, both glutamatergic and GABAergic WT-expressing cells from *Mecp2e1*$^{-/+}$ showed a large number of significant

downregulated genes (blue) but a low number of upregulated genes (red), despite these cell populations being WT-expressing. These differences in gene expression were likely due to non-cell-autonomous effects of the *Mecp2e1* mutation on nearby WT-expressing cells. Further evidence was obtained from the experimental comparison from experiment 5, where mutant-expressing glutamatergic and GABAergic neurons from female *Mecp2e1*$^{-/+}$ were compared to WT-expressing cells from female *Mecp2e1*$^{-/+}$, resulting in only 10 DEGs, compared to 2,216 in experiment 3 (non-cell-autonomous WT vs WT) (Table 1). This non-cell-autonomous effect was dynamic over time, as glutamatergic cells showed mostly upregulated genes with only a few downregulated genes at P60, while GABAergic cells only showed upregulated genes (Fig. 4d). Interestingly, at the late disease stage P150, the number of dysregulated genes were diminished and primarily back to being downregulated, indicating a dynamic process of non-cell-autonomous effects across disease progression. In order to test this hypothesis, we overlapped the significant (Limma-VoomCC adjusted p-value ≤ 0.05) DEGs from each cell type and time point (Fig. 4e, f). In glutamatergic cells, the largest overlap (129 DEGs) was between P30 and P60 and only 3 DEGs were in common to all three time points (Fig. 4e). Similar

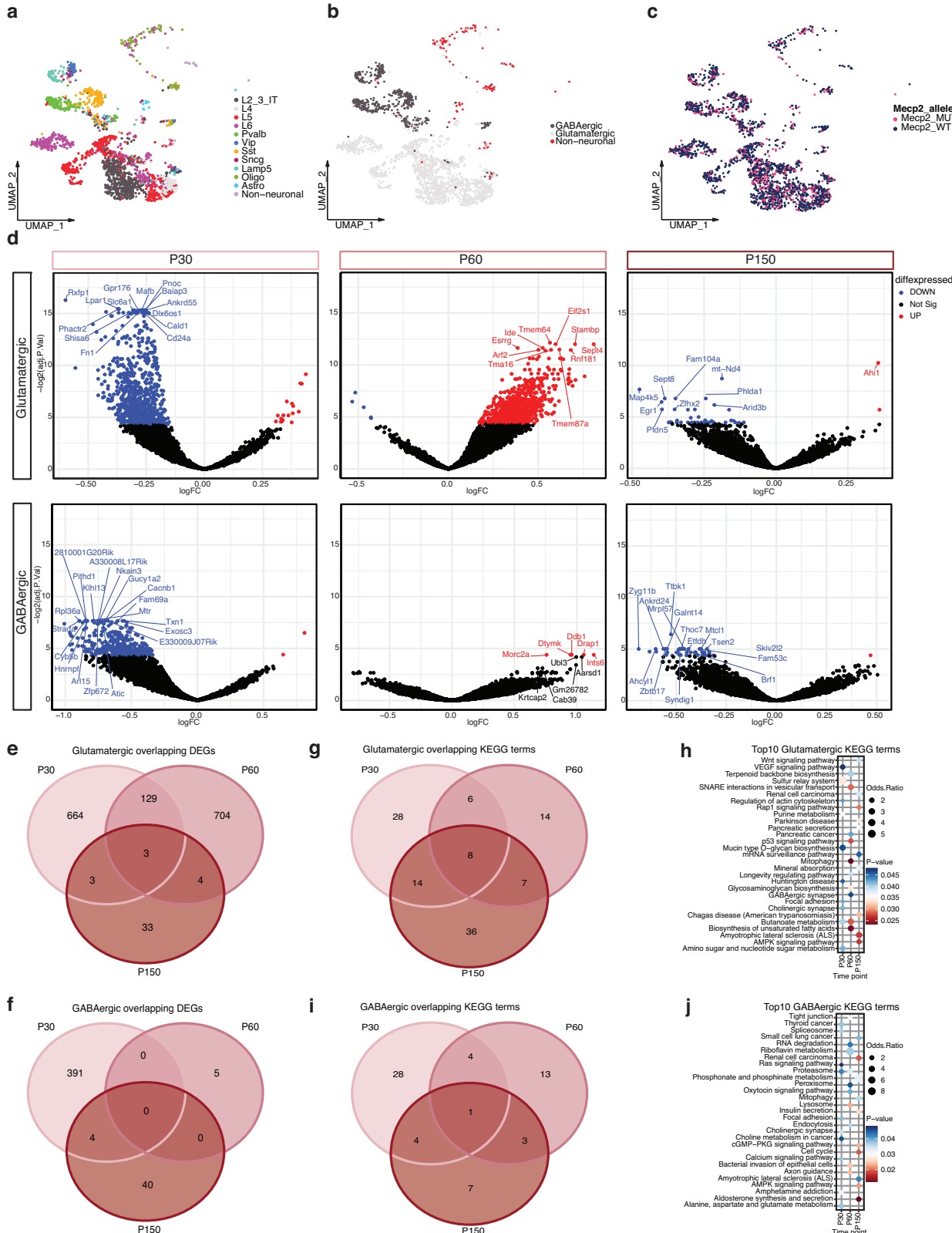

**Fig. 4 | Dynamic non-cell-autonomous effects on differentially expressed genes and KEGG pathways over disease progression.** As shown in Experiment #3 (Fig. 1d), we compared WT cells from *Mecp2e1*$^{-/+}$ female with WT cells from *Mecp2e1*$^{+/+}$ glutamatergic and GABAergic neurons longitudinally. **a** UMAP plot of cell types identified in the mosaic females. **b** UMAP plot of the female cortices showing the clustering of the broad cell type categories. **c** UMAP plot of mosaic female cells parsed by *Mecp2* allele **d** Volcano plots showing differentially expressed

genes (DEGs) of the mouse cortical neurons contrasting WT cells from WT *Mecp2e1*$^{+/+}$ females and WT cells from *Mecp2e1*$^{-/+}$ mosaic females. **e, f** Venn diagrams of overlapping glutamatergic and GABAergic DEGs respectfully over time. **g, i** Venn diagrams of significant KEGG terms of glutamatergic and GABAergic neurons over time. **h**. Top 10 KEGG terms of glutamatergic neurons over time. **j**. Top 10 KEGG terms of GABAergic neurons over time.

results were seen in GABAergic neurons, where DEGs were predominantly unique to each time point (Fig. 4f).

In order to look for functional pathway enrichments of non-cell-autonomous effects of *Mecp2e1*[-/+] mosaicism, KEGG analysis on significant DEGs (p-value ≤ 0.05 from glutamatergic and GABAergic neurons was performed and significant (adjusted p-value ≤ 0.05) terms overlapped across time (Fig. 4g, i). Non-cell-autonomous DEGs from glutamatergic cells were enriched for 8 terms that were shared across all disease stages which include Parkinson, Alzheimer, and Huntington diseases, as well as homeostatic pathways of retrograde endocannabinoid signaling, ubiquitin mediated proteolysis, oxidative phosphorylation, and protein processing in endoplasmic reticulum, while the P60 time point was uniquely enriched for terms such as GABAergic synapse and SNARE interactions in vesicular transport (Fig. 4h). Further, glutamatergic cells showed molecular dysregulation associated with MeCP2 activity such as mRNA surveillance pathway, cholinergic synapse, and AMPK signaling pathway (Fig. 4h). In contrast, GABAergic cells shared axon guidance as an enriched pathway common across all time points (Fig. 4i). Other interesting RTT related pathways included metabolism and energy related terms such as riboflavin metabolism, phosphonate and phosphonate metabolism, choline metabolism, and alanine, aspartate, and glutamate metabolism (Fig. 4j).

In order to compare these non-cell-autonomous effects to cell-autonomous effects over the disease progression in a cell category specific manner, we followed the fourth experiment design (Fig. 1d) and compared the MUT *Mecp2e1* expressing cells from the *Mecp2e1*[-/+] mosaic females to the WT *Mecp2* expressing cells from the *Mecp2e1*[+/+] females (Supplementary Fig. 10a). Similar to the results of experiment 3, glutamatergic and GABAergic significant DEGs were predominantly time point specific (Supplementary Fig. 10b, c). Glutamatergic cells had 12 significant KEGG pathways shared over time while GABAergic cells had 6 significant terms both containing RTT related pathways such as mRNA surveillance and circadian rhythm (Supplementary Fig. 10d–g). In order to examine if the dysregulated KEGG pathways are shared between experiment 3 and experiment 4, a comprehensive overlap test was performed showing that the majority of the pathways are unique to each experiment and each time, with the glutamatergic and GABAergic P150 KEGG pathways from non-cell-autonomous DEGs outnumbering those of cell-autonomous (17 in exp 3 vs 1 in exp 4) (Supplementary Fig. 11a, b).

Lastly, we examined non-cell-autonomous effects by comparing MUT-expressing to WT-expressing cells within the mosaic *Mecp2e1*[-/+] females, as described in experiment 5 (Fig. 1d). Overall, glutamatergic and GABAergic neurons had only a few genes dysregulated which were mostly at P150 when analyzed separately (Supplementary Fig. 12a). For higher statistical power in KEGG term enrichment, DEGs glutamatergic and GABAergic cells were each combined across time points, revealing dysregulated retrograde endocannabinoid signaling and other pathways (Supplementary Fig. 12b, c). The top10 enriched KEGG pathways when both glutamatergic and GABAergic cells were combined across all time points included pathways involved in cell signaling and addiction (Supplementary Fig. 12d). The differences between WT-expressing and MUT-expressing cells within mosaic females in experiment 5 were far less than the differences between WT-expressing cells in mosaic *Mecp2e1*[-/+] females compared to WT cells in *Mecp2e1*[+/+] females in experiment 4. Together, these analyses demonstrate that transcriptional dysregulation across disease progression in mosaic *Mecp2e1*[-/+] females is dynamic, disease stage specific and dominated by non-cell-autonomous effects on homeostatic gene pathways.

### Human RTT cortical cell transcriptional dysregulation is recapitulated by the female but not the male RTT mouse model

To examine how closely *Mecp2e1*[-/+] mice phenocopy Rett syndrome (RTT) at the cellular transcriptome level, we examined the relationship between altered transcript levels by cell type in mouse *Mecp2e1* deficient and human *MECP2*[-/+] cortices. Thus, snRNA seq analysis was performed on eight *MECP2*[-/+] (RTT) and eight age matched control female cortex samples from post-mortem human brains (Fig. 5a, Supplementary Data 3). Nine neuronal

and six non-neuronal cell type clusters could be assigned from these human cortices based on 3,000 gene markers from the Bakken Trygve et al. dataset[20] (Fig. 5b). Cell type labeling based on scTransform containing elevated expression of at least three cell marker genes was validated (Fig. 5c). DEG analysis via Limma-VoomCC compared *MECP2*[-/+] to *MECP2*[+/+] cortical cells, resulting in cell type-specific dysregulated genes (Fig. 5d). Importantly, of the top 20 upregulated DEGs identified by Limma-VoomCC at the adjusted p-value ≤ 0.05 level in female *MECP2*[-/+] cortical cells or significant DEGs (adjusted p value ≤ 0.05) in *Mecp2e1*[-/+] female mouse cortices with 14 gene transcripts out of 20 upregulated (Fig. 5e). Similarly, of the top 20 Rett cortical Limma-VoomCC DEGs that were significantly downregulated (adjusted p value ≤ 0.05), the homologous *Mecp2e1*[-/+] female gene transcripts were also downregulated (Fig. 5e). In contrast, there were very few overlapping DEGs between human RTT and *Mecp2e1*[-/y] male cortical cell transcriptomes (Fig. 5f). This demonstrates that *Mecp2e1*[-/+] female mice are a better model for the dynamic transcriptomic dysregulation due to cellular complexities in Rett syndrome disease progression.

## Discussion

This study advances our understanding of RTT, offering insights into sex-specific, cell type-dependent, and disease stage-associated transcriptional dysregulation resulting from the cellular complexities related to the X-linked dominant inheritance of *MECP2/Mecp2* mutation. This longitudinal analysis of single cortical cell transcriptomes during the gradual progression of disease symptoms in the *Mecp2e1*[-/+] mouse model of RTT provided several new findings critical to the understanding and treatment of human RTT. First, we demonstrated that the female *Mecp2e1*[-/+] mice are inherently different, not simply less severe, in their transcriptional dysregulation compared to mutant *Mecp2*[-/y] males that completely lack *Mecp2e1*. Second, we identified transcriptionally dysregulated gene pathways across cell types in female *Mecp2e1*[-/+] cortices that were significantly associated with the progression of multiple disease phenotypes over time. Third, we showed that non-cell-autonomous effects in mosaic female *Mecp2e1*[-/+] mice are responsible for the homeostatic gene pathway dysregulations observed dynamically over time. Lastly, and most important for translational relevance, we showed that female mosaic *Mecp2e1* mutant mice better recapitulate the transcriptional dysregulation observed in human RTT cortical cells than *Mecp2* null males and may help explain the complexities of progressive and regressive stages of disease in RTT girls.

The earliest studies examining the effect of MeCP2 levels on transcription in the brain relied on bulk RNA-seq comparing male *Mecp2* null to wild-type controls yielded few differentially expressed gene (DEG) transcripts[21–24]. However, bulk analyses of transcript levels in *Mecp2* null compared to *Mecp2* duplication mouse brain revealed 2582 altered DEGs in hypothalamus[25], 1180 DEG transcripts in cerebellum[26], and 1060 DEGs in amygdala[27]. Interestingly, analysis of transcripts in individual brain cell types yielded non-overlapping lists of DEGs suggesting that bulk tissue DEG analysis suffers from a "dilution effect" potentially masking DEGs[28]. While these initial studies comparing *Mecp2* null to wild-type and *Mecp2* duplication control male brains can reveal gene targets of MeCP2 in vivo, *Mecp2*[-/+] female mice are the relevant model for understanding RTT, where brain cell-autonomous and non-autonomous effects require analysis of individual cells and cell types.

For autosomal genes, heterozygous mutations are expected to show reduced phenotypic severity than the homozygous state, but for X-linked genes, there is the added complication of random XCI that creates epigenetic mosaicism within cell populations. We were able to utilize snRNA-seq 5' in the *Mecp2e1* mouse model to parse by both cortical cell type and mutation to improve understanding of transcriptional dysregulation in RTT. Our results can both help confirm certain aspects of previous bulk transcriptomic studies and help explain some of the prior discrepancies between bulk transcriptomic studies in RTT mouse models. A study using bulk RNA-seq on 7-week-old *Mecp2* null mice showed 48 genes upregulated and 32 genes down-regulated in pathways such as circadian entrainment that are consistent with our single nucleus data, despite the lower overall number of

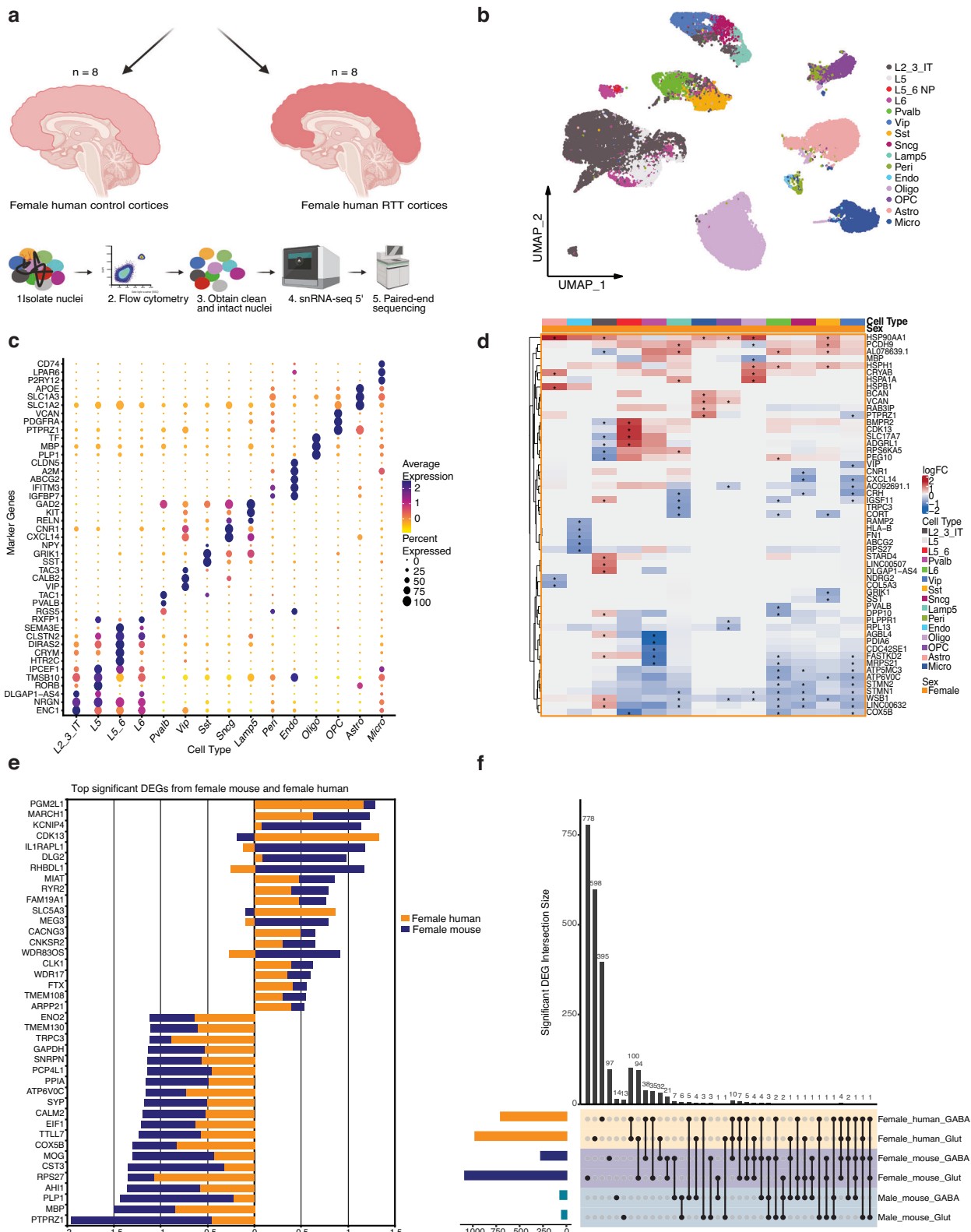

**Fig. 5 | Human RTT cortical neurons share transcriptional dysregulation specifically with *Mecp2e1*[-/+] mosaic female mice. a** A schematic of postmortem human RTT cortices and age/sex matched control cortices. Figure 1a was made using Biorender. **b** UMAP of the unsupervised clustering of cell types identified in the human cortices (*n* = 39336 cells post QC). Cell type labels were transferred from Bakken Trygve et al. 2021[20]. **c** Top gene markers for each cell type in the human cortex. **d** Heatmap of top differentially expressed genes (DEGs) for human female cortices. *indicates adjusted p-value ≤ 0.05. **e** Bar graph showing overlapping of the top significant upregulated and downregulated genes by logFC in female mouse and female human. **f** Upset plot showing overlap of the significant DEGs from both GABAergic and glutamatergic neurons in female human, female mouse, and male mouse.

DEGs identified[29]. We identified circadian entrainment as a homeostatic gene pathway dynamically dysregulated in Pvalb and Sst at P30, L5 excitatory neurons at P60, and astrocytes at P150 in *Mecp2e1*[−/+] cortices. To date, there has been only one prior study conducted using single nucleus RNA-seq in both RTT brain and a mouse model. Renthal et al. used *Mecp2* null cortex at a single time point (8 weeks for males, 12-20 weeks for females) and compared to human RTT cortex with *MECP2* 255X[16]. Our studies were consistent in finding both up- and down-regulated genes with MeCP2 deficiency across cell types and in finding evidence for non-cell-autonomous gene dysregulation, but inconsistent in demonstrating a significant effect of MeCP2 deficiency on repression of long genes. However, differences between the study designs, including genetic mouse model, time points, statistical approaches for DEGs, and single cell technology (iDrops versus 5' V2 technology) could explain the discrepancies. We specifically designed the current study to overcome some of the prior technical limitations, including improvement in signal to noise ratio[30], higher number of genes detected per nucleus, higher UMI per nucleus, and higher number of cells analyzed[31]. Furthermore, we used 3,000 marker genes from the Allen brain atlas cortex single nucleus dataset to label cell types (compared to one marker gene per cell) and used five different statistical approaches to robustly identify differentially expressed genes.

To understand how transcriptional dysregulation in RTT cortex is related to symptom progression, our study uniquely utilized a longitudinal study design and systems biology approaches to correlate networks of dysregulated gene expression patterns with disease phenotypes over time. Remarkably, these disease-relevant gene networks were not specific to individual cell types, but instead were enriched in pathways also dysregulated in neurodegenerative disorders and addiction pathways that regulate brain homeostasis across cell types, including metabolism, circadian entrainment, and retrograde endocannabinoid signaling. Previous studies had shown a link between MeCP2 and addiction[32] that involve *Arc* and *Junb* transcription, consistent with our results in *Mecp2e1*[−/y] cortical cells[33] and KEGG pathways enriched in *Mecp2e1*[−/+] cortical cells. A prior study showing that selective deletion of *Mecp2* from excitatory neurons had no effect on excitatory transmission but reduced inhibitory synapse numbers and neurotransmission in the somatosensory and prefrontal cortex[34] is consistent with our results showing a spread of dysregulated gene pathways from excitatory to inhibitory neurons. A more recent study investigating neurons and astrocytes found KEGG pathways such as calcium signaling pathway and Rap1 signaling pathway were enriched in RTT, consistent with our findings[35].

While non-cell autonomous effects have been previously noted in RTT mouse models, our comprehensive analyses of cellular transcriptomes over disease progression implicates these effects as a central and defining feature of transcriptional dysregulation in RTT mosaic females. Sun et al. argue that the abnormal morphologies of neurons and astrocytes in human RTT are caused by non-cell-autonomous effects driven by altered gene expression and enriched energy related KEGG pathways, consistent with our findings from experiment 2[35]. Defects in signaling pathways suggests RTT disease progression is not caused exclusively by autonomous transcriptional changes in individual cells, but rather due to a failure of wild-type *MECP2* expressing cells to compensate for mutant *MECP2* expressing cells.

Since RTT in humans almost exclusively affects females, our results have important implications for translational medicine. First, pre-clinical models for testing new therapies should be female and construct-relevant, ideally modeling actual human RTT mutations. While male *Mecp2* null models provide important basic insights into MeCP2 function, we clearly demonstrate that the *Mecp2e1* deficient males do not recapitulate the transcriptional dysregulation observed in RTT human cortical cells as well as their female mutant littermates. Furthermore, the non-cell-autonomous dynamic waves of dysregulation in WT-expressing cortical neurons may help explain why human RTT symptoms appear gradually and are staggered in a series of regressions followed by plateaus. Our results showing that transcriptional dysregulation appears pre-symptomatic in female *Mecp2e1*

mutant across multiple cortical cell types suggest that diagnosis and treatment should ideally begin as early as possible, potentially by including *MECP2* in newborn screening panels. To date, the only drug in the market for RTT is Trofinetide which is based on IGF-1 a growth factor previously used for diseases such as Laron syndrome and liver cirrhosis[36,37]. The overlap with other neurologic disease pathways including oxidative phosphorylation suggests that some existing drugs for neurodegenerative disorders could potentially be repurposed to counteract some of the RTT non-cell-autonomous transcriptional dysregulations in pathways regulating homeostasis. Conversely, the molecular pathogenesis of RTT may provide insights for understanding epigenetic regulation of transcriptional homeostasis of gene pathways relevant to common neurodegenerative and addiction disorders.

## Methods

### *Mecp2-e1* mouse breeding and phenotyping
*Mecp2e1*[−/+] female dams were bred with C57BL6/J wild-type male mice (Jackson Labs strain 000664) to produce *Mecp2e1*[−/+] *Mecp2e1*[+/+], *Mecp2*[−/y] and *Mecp2*[+/y] littermates for single-nuclei RNA seq. Mice were weaned at 21 days post-natal and genotyped using DNA from tail snips as described[4]. Mouse disease scoring was performed blinded based on a seven-point scale with points assigned for matted fur, unusually large abdomen, skin lesions, hypoactivity and responses to tail suspension[4]. All animal procedures were approved by the UC Davis Institutional Animal Use and Care Committee (IACUC). We have complied with all relevant ethical regulations for animal use.

### Single nuclei isolation for mouse and human post-mortem cortex
*Mecp2*-e1 and control mice were sacrificed by carbon dioxide inhalation just prior to brain removal. Cerebral cortex was removed from each brain from the mice. Human cortical samples were obtained from National Institute of Child Health and Human Development (NICHD) Brain and Tissue Bank for Developmental Disorders at the University of Maryland, a public biorepository of deidentified deceased donors. All ethical regulations relevant to the use of these samples were followed. About 10 μg of cerebral cortex tissue was isolated from human post-mortem and control samples. Single nuclei were prepared from the left hemisphere cortex according to a previously established protocol Martelotto (https://cdn.10xgenomics.com/image/upload/v1660261285/support-documents/CG000124_Demonstrated_Protocol_Nuclei_isolation_RevF.pdf). Briefly, a 3.0 mm² section of cortex was removed from each mouse brain. Both mouse and human brain tissue were minced with a scalpel then homogenized in 0.5 ml of nuclei lysis buffer with RNAse inhibitor (Roche, Indianapolis, ID) then transferred to a larger tube with an additional 1.0 ml of nuclei lysis buffer, mixed then incubated on ice for 5 min. Nuclei were filtered from the lysate using a 70 μM FlowMi cell strainer (Sp-Belart, Wayne, NJ). Nuclei were pelleted at 4 °C for 5 min at 500×G, resuspended in 1.5 ml of nuclei wash buffer, incubated for 5 min. Nuclei were then pelleted again as above then washed twice in nuclei wash and resuspension buffer then filtered with a 35 μM FlowMi filter (Sp-Belart, Wayne, NJ) then resuspended in nuclei wash and resuspension buffer with 5 μgs/ml DAPI and assayed on a Countess cell counter to determine concentration and nuclear integrity (Fisher Scientific, Waltham, MA). Nuclei were then sorted to remove debris and nuclear aggregates on a MoFlow Astrios cell sorter (Beckman-Coulter, Brea, CA). Approximately,150,000 nuclei per sample were sorted and stored on ice prior to snRNA-seq 5' library generation.

### Single nuclei-RNA sequencing
Single Cell 5' Library & Gel Bead Kits (10× Genomics, Pleasanton, CA) were used to prepare cDNA and generate bar coded and indexed snRNA-seq 5' libraries according to the manufacturers protocol. 10,000 nuclei per sample were targeted. snRNA-seq 5' libraries were balanced using a Kapa library quantification kit (Roche, Indianapolis, IN) and pooled to generate 150 base pair, paired end sequences from using a NovaSeq S4 sequencer (Illumina,

San Diego, CA). Mouse cortices had about 75,000 reads per cell on average and 240,437,728 reads per sample on average. Human cortices had about 50,000 reads per cell on average and 300,000,000 reads per sample on average.

## Pre-processing and quality control

Cellranger v.2.0.2 was used to align the mouse raw reads to mm10-1.2.0 reference genome and the human raw reads to GRCh38 human reference genome. Cell by gene count matrices were used to create a Seurat object using Seurat_4.3.0.1 in R 4.2.2. Mouse samples were filtered with the criteria that cells should have less than 7% mitochondrial, greater than 200 and less than 5,625 genes and greater than 208 and less than 16,300 UMI respectively.

## Cell type identification by dimensionality reduction

The expression counts were log transformed and normalized via Seurat 4.3.0.1. Information about the samples such as sex, genotype, time point, disease score, body weight and *Mecp2e1* expression allele were all added to the metadata. Single cell mouse and human cortex data from the Allen brain institute were used as a reference for cell type labeling both data sets separately[11,38]. scTransform was used to align cell types and transfer labels over to the Rett data. Cell marker test was performed for validating the cell type labeling. Dot plots showing validation of the cell type markers were created via scCustomize 2.1.1 (10.5281/zenodo.5706430).

## DEG analysis

A total of five different DEG analysis methods were used to evaluate the best method for comparing mutant samples to WT samples in a cell-type-specific manner. EdgeR, Limma, and DeSeq2 yielded inconsistent DEGs (Supplementary Fig. 1). For experiments 1 and 2, low expressing genes were filtered out. Low expressing was defined by expression in less than 25% of cells of a given cell type. LimmaVoomCC was used on the remaining high expressing genes to determine differentially expressed genes while considering cells from the same mouse will have correlated expression. For the low expressing genes, DEsingle was used for DEG analysis on genes that are not as robustly expressed (expressed in <25% of cells of type). For experiments 3, 4, and 5, LimmaVoom was exclusively used to identify differentially expressed genes. For each of the DEG experiments, the number of cells were normalized by down sampling. Parameters for all DEG analysis are available in the GitHub repository.

## KEGG analysis

DEGs with a p-value of ≤0.05 from each of the experiments were used as the input for KEGG analysis. This was performed using the R package enrichR 3.2.

The 20 upregulated and downregulated genes were determined by Limma-VoomCC adjusted $p$ value < 0.05. We also included gene ontology analysis using the same DEGs.

## hdWGCNA analysis

Cells from both males and females in the processed Seurat object were used as the input for hdWGCNA analysis. We also included phenotype data such as disease score. The criteria for the fraction of cells that a gene needs to be expressed in order to be included was set at 5%. The network type used is signed with a softpower of 0.8. A total of 9 modules were produced and scores for each module was computed using UCell method. Standard pipeline for hdWGCNA 0.2.4 were followed and the parameters are available in the GitHub repository.

## WT and mutant cell parsing in mosaic female mouse cortices

All *Mecp2* reads were extracted from the raw fastq files generated from each individual sample. abBLAST 3.0 and BWA 0.7.17 mem were used in conjunction to extract *Mecp2* reads (alleler.py). The reference used for alignment was 100 bp of the *Mecp2* gene; 50 bp upstream of the exon1 start codon and 50 bp downstream. With the aligned reads, the number of mutant (TTG) and wild type (ATG) start codons were counted using alleler.py. Each read also contains the cell barcode and UMI information which was used to add the mutant cell and wild-type cell information back to the Seurat object as metadata.

## Overlap of human and mouse DEGs and KEGG pathways

Limma-VoomCC DEGs from both human cell types and mouse cell types were filtered at adjusted $p$-value ≤ 0.05. Significant human DEGs were overlapped with female mouse and male mouse respectively. GeneOverlap 1.38.0 package was used to perform a Fisher's exact test to determine the significance of overlapped genes. The same overlap approach was performed to determine the significant overlapping mouse and human KEGG pathways.

## Statistics and reproducibility

There were 28 mouse cortical samples were collected at four different timepoints corresponding to three different disease stages ($n$ = 28). Four different Mecp2e1 genotypes were considered that include both sexes producing *Mecp2e1$^{-/+}$ Mecp2e1$^{+/+}$, Mecp2$^{-/y}$* and *Mecp2$^{+/y}$* littermates. At the P30 and P60 timepoints, there were two biological replicates of the WT and mutant animals for each sex. At P120, there were two WT and mutant male replicates while at P150, there were four WT and mutant replicates. For DEG analysis, for each gene in each sample, the DEG analysis utilizes the summarized count which follows a negative binomial distribution with mean equal to the multiplication of library size and relative abundance (the gene expression levels), and the variance for each gene is a function of the mean. Significant DEGs are called based on adjusted p value of ≤0.05 which utilizes the Benjamini-Hochberg method to adjust for false discovery. Significant DEGs were used as the input for the KEGG pathway analysis. KEGG analysis uses fishers exact test to determine if the overlap of input DEGs with the background genes are significant (adjusted $p$ value of ≤0.05). Overlap tests of human and mouse DEGs and KEGG pathways were both performed using GeneOverlap which also utilizes a Fisher's exact test to determine if the overlap of two gene sets is significant. hdWGCNA used Pearson correlation test to make comparisons of gene expression with the phenotypes. Significant correlations were determined based on adjusted $p$ value of ≤0.05.

## Reporting summary

Further information on research design is available in the Nature Portfolio Reporting Summary linked to this article.

## Data availability

Data are available through NCBI accession number PRJNA1142998 and PRJNA1157887.

## Code availability

The analysis pipeline for the study is available[39]: https://github.com/osmansharifi/snRNA-seq-pipeline https://doi.org/10.5281/zenodo.13761244.

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

## Acknowledgements

We would like to thank Bridget McLaughlin, for her expertise in flow cytometry analysis and cell sorting, Lutz Froenicke, and Diana Burkart-Waco, for their expertise with 10X genomics snRNA-seq protocols. NIH NIAA grant 1R01AA027075 to Janine M. LaSalle, and NIH Shared Instrumentation Grant 1S10OD010786-01 to the UC Davis DNA technologies core. This project was supported by the University of California Davis Flow Cytometry Shared Resource Laboratory with funding from the NCI P30 CA093373 (Comprehensive Cancer Center), and S10 OD018223 (Astrios Cell Sorter) grants, with technical assistance from Bridget McLaughlin and Jonathan Van Dyke. We also thank assistance by center grants from the National Institute of Child Health and Human Development (P50 HD103526) and the National Institute for Environmental Health Sciences (P30 ES023513).

## Author contributions

O.S. conducted the study, analyzed and interpreted data, and drafted the manuscript. V.H., K.N., K.F., I.K. participated in data analysis. S.H. and D.Y. performed the mouse breeding and sample collection. N.J. and G.Q. provided insight into snRNA-seq data. D.Y. and J.L. conceptualized the study, designed experiments, interpreted data and drafted and revised the manuscript.

## Competing interests

The authors declare no competing interests.
