## [Transparent Peer Review file · Communications Biology]

Sex-specific single cell-level transcriptomic signatures of Rett syndrome disease progression

Corresponding Author: Dr Janine LaSalle

Version 0:

Reviewer comments:

Reviewer #1

(Remarks to the Author)

This manuscript by Sharifi et al reports their 5' snRNA-seq analysis of mouse cerebral cortex of a Rett syndrome mouse model (Mecp2e1 mutant model). The authors collected data from a unique and more patient relevant mouse model and performed a comprehensive comparative analysis to study the effects of cellular mosaicism, sex, cell types, and disease stages. The most interesting finding is related to sex difference, as the authors emphasized. The study and findings are very important and could represent significant advance. Although the bulk of study is bioinformatics analysis of snRNA-seq data, the analytic plan is well designed and comprehensive and. Obviously some sort of validation will further enhance the report, but it is up to the authors. The major issue and thus limitation, however, is the lack of biological replicates. There seems to be only 1 sample per sex per genotype per stage, except at P150 probably. The authors should specifically mention this in the manuscript. Yes, single cell (single nuclei) transcriptomics technology is great but it has a key limitation in the statistical detection of differentially expressed genes (DEGs) between groups, i.e., inflated statistics and power (e.g., PMID 34584091). The authors did a benchmark comparison of five statistical methods and eventually chose to use Limma-VoomCC. This is greatly appreciated, but none of these five would solve the problem, and it has been reported that pseudobulking is a much robust approach. Unfortunately this will need replicates, which the authors didn't have. With only 1 sample per comparison group, it is almost impossible to evaluate if the observed sex difference is a reproducible finding. Using DEG numbers as a metric is not a good idea. There is not a good way to get around this issue, but perhaps the authors could do some testing, for example, splitting the cells of a sample into two halves, run DEG analysis between them to get a baseline false positive, separately for the male and female samples. Alternatively, maybe try metacell approaches by grouping cells into metacells and then do DEG analysis. Nevertheless, for the human samples, the authors have a few replicates and can perform a pseudobulk-based DEG analysis and compare the DEGs with those from their current method. If these issues are addressed and resolved, all the downstream bioinformatics analysis and results would stand.

Two more comments, what is the current consensus why Rett syndrome is rare in male? The authors need to deposit their data to GEO.

Reviewer #2

(Remarks to the Author)

In this study, Osman Sharifi and colleagues conducted a longitudinal single-nucleus RNA sequencing (snRNA-seq) analysis on the cerebral cortex of a construct-relevant Mecp2e1 mutant mouse model of Rett syndrome (RTT) at three postnatal time points, including both male and female mice. They designed five comparison groups to investigate the transcriptional effects of sex, cell type, and mosaicism caused by random X-chromosome inactivation (XCI) on progressive disease phenotypes. Additionally, they included snRNA-seq data from human MECP2-/+ cortices, discovering that the top 20 differentially expressed genes (DEGs) in human samples were also significant DEGs in Mecp2e1-/+ female mice, with few overlapping DEGs found in Mecp2e1-/- male mice. Their scRNA-seq analysis provides insights into sex-specific, cell type-dependent, and disease stage-associated transcriptional changes arising from the cellular complexities of X-linked dominant inheritance of MECP2 mutations. While the study is well-designed and meaningful, several points require improvement, correction, or clarification before further consideration for acceptance.

1. The reasons on selecting Limma-VoomCC for DEG analysis are not convincing. The authors only showed the venngrams for comparing significant DEGs using Limma, EdgeR and DESeq2 from the male sn-RNA seq data. There were no data for

their final choice Limma-VoomCC, or the comparisons for other experiments. The authors should consider provide further evidence to support their choice.

2. One of this paper's highlights is to reveal non-cell-autonomous transcriptional dysfunction by parsing WT-expressing and Mecp2e1 mutation-expressing cells. The authors should check the contributions from each time point to the WT/Mutant Mecp2 expressing cells from Mecp2e1-/- female and Mecp2e1-/-y snRNA-seq data in Supplemental Figure 9. Another way is to plot cell clustering based on time points, like in Fig. 4a-c. Also, the cell numbers shown in Line 242-243 are not consistent with Supplemental Figure 9.

3. While it is appreciated that the authors comparing the mice data with human RTT cortices, the authors should disclose more information for these eight MECP2-/+ RTT samples, including specific genotypes (mutation sites), ages, and whether the genotyping for these samples are suitable for paring WT/MUT Mecp2 expressing cells in human, if doable, which would be very interesting.

4. The description on the numbers of cell types in Line 84-86 was inconsistent with Fig. 1c.

5. The citation in Line 110 should include "Table 1", for this part is the description for it. And there are 165, not 169, higher DEGs shown in male mice. Also, the total significant DEGs number in supplemental table 1 is 1440, but not 1436 shown in Line 109. Again, the number "862 in experiment" in Line 266 should be "2126". The authors should double check the inconsistencies.

6. The description in Line 182 "pathways dysregulated in the female mouse model" should be "pathways dysregulated in the male mouse model" ?

7. The authors should explain what the disease score is, and how they quantify it.

8. The description in Line 275 is not consistent with Figure 4e.

9. The authors should explain the criteria for selecting the top 20 upregulated or downregulated genes, based on adjusted p values or folding change values. There seems only 16 upregulated human genes in Fig. 5e, not 20 as stated.

10. There are at least two duplicate citations shown in the references: the 4th and 10th and the 12th and 39th.

Version 1:

Reviewer comments:

Reviewer #1

(Remarks to the Author)

The authors have addressed my previous comments to the extent they could do. It is unclear if any of the DEG analysis methods was conducted using "pseudobulk" method. This is possible since the authors have 2 (or 4) replicates, though no ideal, to do either a 2 vs 2 or 4 vs 4 comparisons when applicable. I'd suggest the authors to clarify this before submitting their final version.

Reviewer #2

(Remarks to the Author)

The authors have addressed my concerns and corrected the inconsistency in the original manuscript. I have no further questions.

We'd like to thank the editor and reviewers for their time in a constructive review of our manuscript. We were pleased with the overall positive comments, including "comprehensive," "very important," "significant advance" "well-designed" and "meaningful". The revised manuscript has been significantly improved in response to the reviewer's critiques, including new data analyses included in Supplemental Figures 1, 2, 9 and Supplemental Table 3. Responses to reviewers are highlighted in red and changes to the manuscript are highlighted in blue below, and using track changes in the revised manuscript.

Reviewers' comments:

Reviewer #1 (Remarks to the Author):

1. This manuscript by Sharifi et al reports their 5' snRNA-seq analysis of mouse cerebral cortex of a Rett syndrome mouse model (Mecp2e1 mutant model). The authors collected data from a unique and more patient relevant mouse model and performed a comprehensive comparative analysis to study the effects of cellular mosaicism, sex, cell types, and disease stages. The most interesting finding is related to sex difference, as the authors emphasized. The study and findings are very important and could represent significant advance. Although the bulk of study is bioinformatics analysis of snRNA-seq data, the analytic plan is well designed and comprehensive and. Obviously some sort of validation will further enhance the report, but it is up to the authors. The major issue and thus limitation, however, is the lack of biological replicates. There seems to be only 1 sample per sex per genotype per stage, except at P150 probably. The authors should specifically mention this in the manuscript.

Thank you for taking the time to review and provide feedback for our manuscript. As suggested, validating our transcriptomic data would be valuable; however, due to the nature of the single cell technology, validating through traditional methods such as qRT-PCR is not feasible for validation of genes at the single cell level. We did validate our datasets by comparing our DEGs between mouse and human cortical cells (Figure 5). We apologize for the confusion regarding biological replicate representation. We have a minimum of 2 replicates per sex per genotype per timepoint and 4 replicates per genotype at P150. Information on biological replicates is included in Table 1 and now better explained in the revised Results section.

Results (p. 2, lines 82-85)

A total of 28 cortex samples were used, including two per timepoint and genotype for males (PND30, 60, 120) and two per genotype for females at PND30 and 60), but four replicates per genotype were used in females at P150 in order to study mosaicism (Table 1).

2. Yes, single cell (single nuclei) transcriptomics technology is great but it has a key limitation in the statistical detection of differentially expressed genes (DEGs) between groups, i.e., inflated statistics and power (e.g., PMID 34584091). The authors did a benchmark comparison of five statistical methods and eventually chose to use Limma-VoomCC. This is greatly appreciated, but none of these five would solve the problem, and it has been reported that pseudobulking is a much robust approach. Unfortunately this will need replicates, which the authors didn't have. With only 1 sample per comparison group, it is almost impossible to evaluate if the observed sex difference is a reproducible finding.

Thank you for providing us with the reference paper comparing DEG analysis tools. While Squair et al. show performance of single cell DEG tools in relation to each other, they do not consider inter-correlation relationships of cells within the same animals. Further, they conclude that aggregate cells were the main driving factor for false positive highly expressed DEGs. To mitigate these limitations, we consider inter-cellular correlations of all individual samples by integrating consensus correlation onto LimmaVoom. In addition, we implemented a stringent

filtering criterion to exclude genes that are not expressed in at least 25% of a given cell type which was not implemented by Squair et al. Lastly, we performed DEG analysis using one of the top performing DEG tools suggested by Squair et al. (EdgeR) and added two new panels (Supplemental figure 1a,b) showing comparison with Limma-VoomCC and other methods.

Results (p. 3, lines 110-113)

An overlap of DEGs between EdgeR and Limma-VoomCC showed that Limma-VoomCC identifies mostly the same DEGs, but EdgeR identifies many more DEGs that likely include many false positives, as highlighted by Squair et al. (**Supplemental Figure 1b**).

3. Using DEG numbers as a metric is not a good idea. There is not a good way to get around this issue, but perhaps the authors could do some testing, for example, splitting the cells of a sample into two halves, run DEG analysis between them to get a baseline false positive, separately for the male and female samples. Alternatively, maybe try metacell approaches by grouping cells into metacells and then do DEG analysis. Nevertheless, for the human samples, the authors have a few replicates and can perform a pseudobulk-based DEG analysis and compare the DEGs with those from their current method. If these issues are addressed and resolved, all the downstream bioinformatics analysis and results would stand.

Thank you for the suggesting testing DEG methods and sample splitting to address DEG analysis method limitations. Two new panels were added to Supplemental figure 2e-f to address this question. As suggested, we split our samples into males and females and performed DEG analysis via DEsingle, LimmaVoomCC and EdgeR. As a reminder, EdgeR is one of the methods suggested by Squair et al. While the absolute number of DEGs differ across the three tools, there was a consistent finding of increasing DEGs for males and decreasing DEGs for females temporally that was independent of the method used.

Results (p. 3, lines 133-136)

Analysis of DEGs by Limma-VoomCC, DEsingle, and EdgeR revealed that cell type transcriptional changes associated with *Mecp2e1* deficiency were markedly different by sex and disease stage in multiple cortical cell types and these differences were consistent across bioinformatic methods (**Figure 2, Supplemental Figure 2**).

4. Two more comments, what is the current consensus why Rett syndrome is rare in male? >90% of *MECP2* mutations occur on the paternal X chromosome, thereby affecting females. *MECP2* mutation correlates with early onset of severe disease phenotypes such as seizures that reduce survival in males. We have included more about this in the Introduction.

Introduction (p. 1, lines 41-44)

Rett syndrome most often occurs through spontaneous missense or truncation mutations in the X linked gene *MECP2* that are mostly germline mutations in the X linked gene *MECP2* that are mostly paternal, thereby predominantly affecting XX females². Males with *MECP2* mutations that survive infancy are rare but can be observed clinical phenotypes ranging from severe neonatal encephalopathy to cognitive impairment³.

5. The authors need to deposit their data to GEO.

All data has been uploaded to NCBI and will be publicly available upon publication.

Data Availability (p. 13, line 601)

Data are available through NCBI accession number PRJNA1142998.

Reviewer #2 (Remarks to the Author):

In this study, Osman Sharifi and colleagues conducted a longitudinal single-nucleus RNA sequencing (snRNA-seq) analysis on the cerebral cortex of a construct-relevant *Mecp2e1* mutant mouse model of Rett syndrome (RTT) at three postnatal time points, including both male and female mice. They designed five comparison groups to investigate the transcriptional effects of sex, cell type, and mosaicism caused by random X-chromosome inactivation (XCI) on progressive disease phenotypes. Additionally, they included snRNA-seq data from human *MECP2*^{-/+} cortices, discovering that the top 20 differentially expressed genes (DEGs) in human samples were also significant DEGs in *Mecp2e1*^{-/+} female mice, with few overlapping DEGs found in *Mecp2e1*^{-/y} male mice. Their scRNA-seq analysis provides insights into sex-specific, cell type-dependent, and disease stage-associated transcriptional changes arising from the cellular complexities of X-linked dominant inheritance of *MECP2* mutations. While the study is well-designed and meaningful, several points require improvement, correction, or clarification before further consideration for acceptance.

Thank you for the very helpful comments on our manuscript.

1. The reasons on selecting Limma-VoomCC for DEG analysis are not convincing. The authors only showed the venngrams for comparing significant DEGs using Limma, EdgeR and DESeq2 from the male sn-RNA seq data. There were no data for their final choice Limma-VoomCC, or the comparisons for other experiments. The authors should consider provide further evidence to support their choice.

Supplemental Figures 1 and 2 were both modified to show comparison of Limma-VoomCC with the other 3 methods used in the study. While all methods were consistent in showing sex and timepoint differences, Limma-VoomCC was the only method that controlled for the inter-correlations of cells from the same animals. Please see the revisions made in response to Reviewer 1, points 2 and 3.

2. One of this paper's highlights is to reveal non-cell-autonomous transcriptional dysfunction by parsing WT-expressing and *Mecp2e1* mutation-expressing cells. The authors should check the contributions from each time point to the WT/Mutant *Mecp2* expressing cells from *Mecp2e1*^{-/+} female and *Mecp2e1*^{-/y} snRNA-seq data in Supplemental Figure 9. Another way is to plot cell clustering based on time points, like in Fig. 4a-c. Also, the cell numbers shown in Line 242-243 are not consistent with Supplemental Figure 9.

Thank you for catching this, we have corrected the inadvertent switch of cell numbers.

Results (p. 6, lines 251-253)

... we identified 1,146 *Mecp2* expressing cells, of which 607 could be genotyped as WT-expressing and 539 were expressing the *Mecp2e1* mutation (**Supplemental Figure 9**).

3. While it is appreciated that the authors comparing the mice data with human RTT cortices, the authors should disclose more information for these eight *MECP2*^{-/+} RTT samples, including specific genotypes (mutation sites), ages, and whether the genotyping for these samples are suitable for paring WT/MUT *Mecp2* expressing cells in human, if doable, which would be very interesting.

These data on the human brain samples are provided in the new Supplementary Table 3. Unfortunately, none of the available RTT brain samples have the same mutation as the mouse model.

Results (p. 8, lines 341-343)

Thus snRNA-seq analysis was performed on eight *MECP2*^{-/+} (RTT) and eight age matched control female cortex samples from post-mortem human brains (**Figure 5a, Supplemental Table 3**).

4. The description on the numbers of cell types in Line 84-86 was inconsistent with Fig. 1c.

Corrected

5. The citation in Line 110 should include "Table 1", for this part is the description for it. And there are 165, not 169, higher DEGs shown in male mice.

Corrected

Also, the total significant DEGs number in supplemental table 1 is 1440, but not 1436 shown in Line 109.

Corrected

Again, the number "862 in experiment" in Line 266 should be "2126". The authors should double check the inconsistencies.

Corrected

6. The description in Line 182 "pathways dysregulated in the female mouse model" should be "pathways dysregulated in the male mouse model" ?

Corrected

7. The authors should explain what the disease score is, and how they quantify it.

We have added this information to the Methods.

Methods (p.11, lines 472-480)

***Mecp2-e1* mouse breeding and phenotyping**

Mecp2e1^{-/-} female dams were bred with C57BL6/J wild-type male mice (Jackson Labs strain 000664) to produce *Mecp2e1^{-/+}*, *Mecp2e1^{+/+}*, *Mecp2^{-/y}* and *Mecp2^{+/y}* littermates for single-nuclei RNA seq. Mice were weaned at 21 days post-natal and genotyped using DNA from tail snips as described⁴. Mouse disease scoring was performed blinded based on a seven-point scale with points assigned for matted fur, unusually large abdomen, skin lesions, hypoactivity and responses to tail suspension⁴. All animal procedures were approved by the UC Davis Institutional Animal Use and Care Committee (IACUC).

8. The description in Line 275 is not consistent with Figure 4e.

Corrected

9. The authors should explain the criteria for selecting the top 20 upregulated or downregulated genes, based on adjusted p values or folding change values.

Methods (p. 12, lines 549-550)

The 20 upregulated and downregulated genes were determined by LimmaVoomCC adjusted *p* value <0.05.

There seems only 16 upregulated human genes in Fig. 5e, not 20 as stated.

Corrected

10. There are at least two duplicate citations shown in the references: the 4th and 10th and the 12th and 39th.

Corrected

Response to Reviewer's comments regards to pseudobulk DEG analysis:

We appreciate the reviewer's recommendation to use edgeR for differential expression analysis. We understand the importance of selecting an appropriate method for DEG analysis in single-cell RNA-seq data. In our study, we added edgeR DEG analysis based on its established suitability for analyzing count data from bulk RNA-seq experiments, as suggested by the reviewer. However, we did not consider pseudobulk analysis methods as an alternative approach. The reasons for not using pseudobulk DEG analysis in our current study are because our primary objective was to investigate the differential expression profiles at the single-cell level rather than aggregating cells into pseudobulk samples. We aimed to capture the heterogeneity and subtle differences in gene expression that can be lost when aggregating cells into pseudobulk samples. While pseudobulk DEG analysis can be powerful, it requires careful consideration of the aggregation strategy and the impact on statistical power. Given the complex nature of our dataset and the specific biological questions addressed, we included edgeR analysis, which is well-suited for handling the count data directly and aligns with the statistical framework of our analysis. The use of edgeR for single-cell RNA-seq data has been documented in recent studies, demonstrating its effectiveness in handling the nuances of single-cell data without the need for pseudobulk aggregation (Nguyen et al. 2023). We aimed to remain consistent with established methodologies that have been validated in similar contexts. Nguyen, H.C.T., Baik, B., Yoon, S. et al. Benchmarking integration of single-cell differential expression. *Nat Commun* 14, 1570 (2023). <https://doi.org/10.1038/s41467-023-37126-3>